# Permafrost-wildfire interactions: Active layer thickness estimates for paired burned and unburned sites in northern high-latitudes

Anna C. Talucci[1], Michael M. Loranty[2], Jean E. Holloway[3], Brendan M. Rogers[1], Heather D. Alexander[4], Natalie Baillargeon[1], Jennifer L. Baltzer[5], Logan T. Berner[6], Amy Breen[7], Leya Brodt[8], Brian Buma[9, 10], Jacqueline Dean[1], Clement J. F. Delcourt[11], Lucas R. Diaz[11], Catherine M. Dieleman[12], Thomas A. Douglas[13], Gerald V. Frost[14], Benjamin V. Gaglioti[15], Rebecca E. Hewitt[16], Teresa Hollingsworth[17,18], M. Torre Jorgenson[19], Mark J. Lara[20], Rachel A. Loehman[21], Michelle C. Mack[22], Kristen L. Manies[23], Christina Minions[1], Susan M. Natali[1], Jonathan A. O'Donnell[24], David Olefeldt[25], Alison K. Paulson[26], Adrian V. Rocha[27], Lisa B. Saperstein[28], Tatiana A. Shestakova[29, 30, 1], Seeta Sistla[31], Oleg Sizov[32], Andrey Soromotin[8], Merritt R. Turetsky[33], Sander Veraverbeke[11], Michelle A. Walvoord[34]

[1] Woodwell Climate Research Center, Falmouth, MA, 02540-1644, USA
[2] Department of Geography, Colgate University, Hamilton, NY, 13346, USA
[3] Department of Geography, Environment and Geomatics, University of Ottawa, Ottawa, K1N 6N5, Canada
[4] College of Forestry, Wildlife, and Environment, Auburn University, Auburn, AL, 36949, USA
[5] Biology Department, Wilfrid Laurier University, Waterloo, ON, N2L 3C5, Canada
[6] School of Informatics, Computing, and Cyber Systems, Northern Arizona University, Flagstaff, AZ, 86011, USA
[7] International Arctic Research Center, University of Alaska Fairbanks, Fairbanks, AK, 99775-7340, USA
[8] Tyumen State University, Tyumen, 625003, Russia
[9] Integrative Biology, University of Colorado (Denver), Boulder, CO, 80304, USA
[10] Environmental Defense Fund, Boulder, CO 80302, USA
[11] Faculty of Science, Vrije Universiteit Amsterdam, Amsterdam, 1081 HV, The Netherlands
[12] School of Environmental Sciences, University of Guelph, Guelph, ON, N3H3Y8, Canada
[13] U.S. Army Cold Regions Research and Engineering Laboratory, Fort Wainwright, AK, 99703, USA
[14] Alaska Biological Research, Inc., Fairbanks, AK, 99708, USA
[15] Water and Environmental Research Center, University of Alaska Fairbanks, Fairbanks, AK, 99775, USA
[16] Department of Environmental Studies, Amherst College, Amherst, MA, 01002, USA
[17] Pacific Northwest Research Station, USDA Forest Service, University of Alaska Fairbanks, Fairbanks, AK, 99708, USA
[18] Aldo Leopold Wilderness Research Institute, Rocky Mountain Research Station, Missoula MT, 59801
[19] Alaska Ecoscience, Fairbanks, AK, 99775, USA
[20] Department(s) of Plant Biology and Geography, University of Illinois Urbana-Champaign, Urbana, IL, 61801, USA
[21] U.S. Geological Survey, Alaska Science Center, Anchorage, AK, 99508, USA
[22] Center for Ecosystem Science and Society and Department of Biological Sciences, Northern Arizona University, Flagstaff, AZ, 86001, USA
[23] U.S. Geological Survey, Moffett Field, 94035, USA
[24] Arctic Network, National Park Service, Anchorage, AK, 99501, USA
[25] Department of Renewable Resources, University of Alberta, Edmonton, AB, T6G 2G7, Canada
[26] Humboldt-Toiyabe National Forest, U.S. Forest Service, Sparks, NV, 89431, USA
[27] Department of Biological Sciences, University of Notre Dame, Notre Dame, IN, 46556, USA
[28] Alaska Regional Office, U.S. Fish and Wildlife Service, Anchorage, AK, 99503, USA
[29] Department of Agricultural and Forest Sciences and Engineering, University of Lleida, Av. Alcalde Rovira Roure 191, Lleida, Catalonia 25198, Spain

[30] Joint Research Unit CTFC–AGROTECNIO–CERCA, Av. Alcalde Rovira Roure 191, Lleida, Catalonia 25198, Spain

[31] Natural Resources Management & Environmental Sciences, Cal Poly, San Luis Obispo, CA, 93401, USA

[32] Oil and Gas Research Institute RAS, Moscow, 119333, Russia

[33] Renewable and Sustainable Energy Institute, Department of Ecology and Evolutionary Biology, University of Colorado Boulder, Boulder, CO, 80309-0552, USA

[34] U.S. Geological Survey, Earth System Processes Division, Denver, CO, 80225, USA

*Correspondence to*: Anna C. Talucci (atalucci@woodwellclimate.org)

**Abstract.** As the northern high latitude permafrost zone experiences accelerated warming, permafrost has become vulnerable to widespread thaw. Simultaneously, wildfire activity across northern boreal forest and Arctic/subarctic tundra regions impact permafrost stability through the combustion of insulating organic matter, vegetation, and post-fire changes in albedo. Efforts to synthesis the impacts of wildfire on permafrost are limited and are typically reliant on antecedent pre-fire conditions. To address this, we created the FireALT dataset by soliciting data contributions that included thaw depth measurements, site conditions, and fire event details with paired measurements at environmentally comparable burned and unburned sites. The solicitation resulted in 52,466 thaw depth measurements from 18 contributors across North America and Russia. Because thaw depths were taken at various times throughout the thawing season, we also estimated end of season active layer thickness (ALT) for each measurement using a modified version of the Stefan equation. Here, we describe our methods for collecting and quality checking the data, estimating ALT, the data structure, strengths and limitations, and future research opportunities. The final dataset includes 48,669 ALT estimates with 32 attributes across 9,446 plots and 157 burned/unburned pairs spanning Canada, Russia, and the United States. The data span fire events from 1900 to 2022 with measurements collected from 2001 to 2023. Time since fire ranges from zero to 114 years. The FireALT dataset addresses a key challenge: the ability to assess impacts of wildfire on ALT when measurements are taken at various times throughout the thaw season depending on the time of field campaigns (typically June through August) by estimating ALT at the end of season maximum. This dataset can be used to address understudied research areas particularly algorithm development, calibration, and validation for evolving process-based models as well as extrapolating across space and time, which could elucidate permafrost-wildfire interactions under accelerated warming across the high northern latitude permafrost zone. The FireALT dataset is available through the Arctic Data Center.

## 1 Introduction

Permafrost, defined as ground that remains at or below 0°C for two or more consecutive years, has become vulnerable to widespread thaw in response to rapid climate warming at high latitudes. Permafrost temperatures have increased over the last 30 years (Romanovsky et al., 2010, Smith et al., 2022, Calvin et al., 2023) resulting in the thickening of the active layer, which is the uppermost, seasonally thawed layer (Harris and Permafrost Subcommittee, Associate Committee on Geotechnical Research, National Research Council of Canada, 1988, Bonnaventure and Lamoureux 2013). Widespread permafrost thaw and

increases in active layer thickness are expected under future climate conditions (Smith and Burgess 2004, Zhang et al., 2008, Derksen et al., 2019, Peng et al., 2023), and these processes are expected to release large amounts of soil carbon to the atmosphere as greenhouse gas emissions (Schaefer et al., 2014, Gasser et al., 2018, Knoblauch et al., 2018, Yokohata et al., 2020, Natali et al., 2021, Schuur et al., 2022, See et al., 2024). Changes to permafrost, particularly near-surface permafrost and the active layer, have important implications for ecology, forestry, hydrology, biogeochemistry, climate feedbacks, engineering, traditional livelihoods, and community safety (Anisimov and Reneva 2006, O'Donnell et al., 2011b, Rocha and Shaver 2011, Bret-Harte et al., 2013, Hugelius et al., 2014, Jones et al., 2015, Li et al., 2019, Turetsky et al., 2020, Gibson et al., 2021, Huang et al., 2024).

Climate change is also intensifying high-latitude wildfire regimes (Kasischke et al., 2010, de Groot et al., 2013, Zhang et al., 2015, Wotton et al., 2017, Hanes et al., 2019, McCarty et al., 2021, Descals et al., 2022, Phillips et al., 2022, Scholten et al., 2022, Zheng et al., 2023, Byrne et al., 2024). Wildfire activity shows interannual variability that is predominantly controlled by subseasonal drying and climate, where prolonged warm and dry conditions in conjunction with fuel accumulation may alter fire regimes and the seasonality of fire (York et al., 2020). The interaction between wildfire and permafrost results in both immediate and long-term effects on the surface energy balance and ground thermal regimes, as well as hydrologic cycling and soil and aquatic biogeochemistry (O'Donnell et al., 2011b, Rocha and Shaver 2011, Bret-Harte et al., 2013, Jones et al., 2015, Li et al., 2019, Hollingsworth et al., 2020, Holloway et al., 2020). These interactions also result in second-order greenhouse gas emissions (O'Donnell et al., 2011c, Jiang et al., 2015, Smith et al., 2015, Jones et al., 2015, Gibson et al., 2018, Li et al., 2019) by making stored soil carbon available for mineralization (O'Donnell et al., 2011c, Rocha and Shaver 2011, Bret-Harte et al., 2013, Hugelius et al., 2014, Jones et al., 2015, Li et al. 2019). Biomass combustion during fires removes the insulating surface vegetation (i.e., moss, lichen, low growing shrubs) and soil organic matter, typically reduces evapotranspiration (Rouse 1976, Amiro 2001, Chambers and Chapin 2002, Chambers et al., 2005, Amiro et al., 2006, Chebykina et al., 2022, Fedorov, 2022), and reduces short-term albedo during thaw season, resulting in increases in the ground heat flux and the expansion of the active layer (Moskalenko 1999, Rocha et al., 2012, Jafarov et al., 2013, Nossov et al., 2013, Jiang et al., 2015, Douglas et al., 2016, Fisher et al., 2016, Gibson et al., 2018). Similarly, tree canopy removal reduces shading in the summer and results in more snow on the ground in the winter, both leading to higher surface soil temperatures and expansion of the active layer into near-surface permafrost, which has been shown across North America (Rocha et al., 2012, Jafarov et al., 2013, Jiang et al., 2015, Zhang et al., 2015, Douglas et al., 2016, Fisher et al., 2016, Gibson et al., 2018) and Eurasia (Moskalenko 1999, Lytkina, 2008, Kirdyanov et al., 2020, Heim et al., 2021, Fedorov, 2022, Petrov et al., 2022). In contrast, across North American Arctic tundra, shrub removal from wildfire results in thinner snow due to increased wind exposure, which causes a reduction of the active layer (Wang et al., 2012, Jones et al., 2024), while Russian scientists note an expansion of the seasonal active layer that is dependent on vegetation communities (Moskalenko 1999, Lytkina, 2008).

Post-fire changes in the energy balance and subsequent increases in the active layer thickness have historically recovered to pre-fire conditions as vegetation succession occurred (Rouse 1976, Amiro 2001, Liu et al., 2005, Amiro et al., 2006), with a maximum active layer thickness often observed 5-10 years post-fire (Rocha et al., 2012, Holloway et al., 2020) but may extend up to 30 or more years post-fire (Gibson et al., 2018, Kirdyanov et al., 2020, Heim et al., 2021). However, this pattern of recovery may be changing alongside climate warming and shifting fire regimes (Brown et al., 2015), and may be further impacted by secondary disturbances (Hayes and Buma, 2021). For example, as wildfire burns across permafrost peatlands, not only is there a thicker and warmer active layer but an expansion of year-round unfrozen ground (i.e., taliks) and thermokarst bogs (Gibson et al., 2018). These changes in active layer thickness and hydrologic dynamics can constrain regeneration by prolonging vegetation recovery and inducing shifts in vegetation composition and structure (Baltzer et al., 2014, Dearborn et al., 2021). Further, near-surface permafrost degradation can lead to ground subsidence, which alters surface hydrology, often leading to water inundation and further degradation (Brown et al., 2015). Where wildfires burn across permafrost landforms (e.g., thermokarst, ice rich areas), deep and irreversible thawing could permanently alter the landscape (Burn and Lewkowicz 1990, Lewkowicz 2007, Sannel and Kuhry 2011, Liljedahl et al., 2016, Rudy et al., 2017, Borge et al., 2017, Mamet et al., 2017, Fraser et al., 2018), releasing long stored soil carbon into the atmosphere (Schuur et al., 2015). Currently, emissions from fire-induced permafrost thaw are underestimated by the scientific community and climate models (Natali et al., 2021, Treharne et al., 2022, Schädel et al., 2024), an issue that is exacerbated by modelling challenges and uncertainties associated with permafrost carbon stocks (Hugelius et al., 2014, Turetsky et al., 2020). The change in active layer thickness over time is a critical diagnostic indicator of permafrost conditions (Brown et al., 2000, Shiklomanov et al., 2010) and a vital component of modelling carbon emissions from fire and non-fire related permafrost thaw.

To provide critical data that can be used for understanding and modelling impacts of wildfire on permafrost, we compiled a dataset of thaw depth measurements from paired burned and unburned sites across the northern high-latitude permafrost zone. This dataset is the first of its kind to focus on paired burned and unburned sites providing a circumpolar/boreal perspective. Climate and ecosystem conditions including drainage, vegetation, and soil characteristics control near-surface permafrost characteristics, and thus in order to detect an influence of wildfire it is necessary to have measurements either pre- and post-fire, or unburned control and burned nearby sites with otherwise similar ecosystem properties. Measuring ALT for paired unburned control and nearby burn sites is more realistic due to the stochasticity of wildfire. Further, unburned control sites provide a benchmark for understanding the impact of wildfire in these dynamic systems. Thaw depth increases over the course of the thawing season until it reaches its maximum depth, i.e., active layer thickness (ALT). This means that early to mid-season measurements do not capture the full depth of the thawed active layer. As such, the variability in thawing season and measurement timing makes it difficult to compare across space and time. Therefore, we standardised thaw depths taken at different times throughout the thawing season, which resulted in an estimated dataset of ALT. Further, capturing the maximum ALT aids in establishing the full scope of permafrost change because it is a critical indicator of thaw dynamics. Depending on the location ALT could occur anywhere from August through November. The overarching goal is to generate a synthesised

data set of ALT for burned/unburned pairs. To achieve this, we had four main objectives for the paper: 1) describe how the data was collected and synthesised for thaw depth measurements of burned sites with paired unburned sites, 2) describe how we standardised thaw depth measurements to end-of-season ALT with estimates of uncertainty, 3) provide details on how to aggregate data to plot, site, and paired burned/unburned means and provide a summary of the data set, and 4) discuss the strengths and limitations of the dataset, along with its potential uses.

## 2 Data and Methods

### 2.1 Data Solicitation and Quality Screening

To assemble a dataset capable of widely characterising the influence of wildfire on permafrost, we solicited field measurements of thaw depth from paired burned and unburned sites from researchers working in boreal forest and tundra ecosystems. Thaw depth refers to depth or thickness of the unfrozen surface soil layer anytime during the thawing season. The data sets that contribute to this synthesis were obtained by measuring depth to refusal using a graduated steel probe (Brown et al., 2000). A steel probe is a typical means of measurements, however, there is potential for error introduced by issues such as identifying the freeze-thaw boundary, soil variability, subsidence, user bias (Brown et al., 2000, Bonnaventure and Lamoureux, 2013, Strand et al., 2021, Scheer et al., 2023). A critical component of the data required an ecologically appropriate unburned site(s) within close proximity that shared similar dominant vegetation, drainage, and climatic conditions to be paired with one or more burned sites, meaning the burned site would have had similar pre-fire conditions to the unburned site. We began by soliciting data from members of the Permafrost Carbon Network and their collaborators and then used literature review to identify additional contributors. Data contributors were required to submit metadata (Table S1) and data via a Google form with required attributes that included their last name, country where data were collected, latitude, longitude, biome, vegetation cover class, site identifier, plot identifier, year data were collected, month data were collected, day data was collected, fire identifier, fire year, whether the site was burned or unburned, organic layer depth, thaw depth, whether the probe hit rocks, whether the depth was greater than the probe, contributors assigned a designation of 'thaw' or 'active' to indicate early-mid or late season measurements respectively, slope, topographic position, pairing, and whether surface water was present. The solicitation resulted in the contribution of 18 datasets with 52,466 thaw depth measurements covering portions of the northern high-latitude permafrost zones in Canada, Russia, and the United States (Table 1, Fig. 1).

**Table 1. Brief description of the data contributions. Table includes the last name of the contributor, geographic location of the data, fire years that were sampled, a brief description of the sampling design and methods (see associated publications for additional detail), and relevant citations associated with the data.**

| Contributor | Country Biome(s) - Location description | Fire years | Sampling design and methods (see publications for additional details) | Citations |
|---|---|---|---|---|

| | | | | |
|---|---|---|---|---|
| Baillargeon | United States Tundra - Yukon Kuskokwim Delta, AK, USA | 1972, 2015 | In 2018, thaw depth was sampled along 30 m transects at 1-m intervals. In 2019, thaw depth was measured along 30 m transects at 2m intervals. We quantified depth to refusal with a tile probe. | Baillargeon et al., 2022 |
| Breen | United States Tundra - Kougarok Fire Complex on the Seward Peninsula, AK, USA | 1971, 1982, 2002, 2011 | Thaw depth was measured in the four plot corners of 1-by-1 m unmarked plots along a chronosequence of time since fire and number of times burned (n=35) and unburned (n=8). Depth to refusal was measured with a tile probe. For each plot, the mean of the 4 depths is reported. | Hollingsworth et al., 2020, 2021 |
| Buma | United States Boreal - Central Alaska black spruce forest | 2004/2005 | Plots randomly placed in the four treatments (unburned, 1 fire in 2004/2005, 2 fires (1970's and 2004/2005), and 3 fires (1950's, 1970's, and 2004/2005). Thaw depth sampled randomly within 1-3 time burn plots (n=5 per plot, 33 plots), measured as depth to active layer at time of sampling (denoted as hitting frozen soil). The maximum depth of the probe was 1.8m. | Hayes and Buma 2021 (design), Buma et al. 2022 unpublished data |
| Delcourt, Veraverbeke | Russia Boreal - Northeast Siberia, Russia | 2018 | In 2019, thaw depth was measured at five evenly spaced locations (every 7.5 m) along a 30 m transect centred within a 30 by 30 m plot. We measured depth to refusal using a pointed, graduated steel rod. Two measurements were taken 1 m apart at each location, totalling 10 measurements per plot. | Diaz et al., 2024, Delcourt et al., 2024 |
| Diaz | United States Tundra -- Alaska, USA | 2022 | Thaw depth was measured using a steel rod probe, which was inserted into the ground to the depth of resistance by the frozen ground. In 20 by 20 m plots, we performed measurements every 2 m. Measurements taken in July-August, one year after the fire. | L.R. Diaz, Vrije Universiteit Amsterdam, unpublished data, 2023 |
| Baltzer, Dieleman, Turetsky | Canada Boreal -- Northwest Territories, Canada | 1940, 1960, 1969, 1971, 1972, 1973, 1980, 1981, 2011, 2013, 2014 | From 2015-2019, thaw depths were measured using a tile probe at 6 locations evenly spaced along a 30 m transect centred within a 30-by-30 m plot. We quantified the depth to refusal. | Dieleman et al., 2022 |
| Douglas, Jorgenson | United States Boreal -- Interior Boreal near Fairbanks, AK, USA | 2005-2020 | Multiple transects visited sporadically over the past ten years. Thaw depths were measured by pushing a metal rod ("thaw probe") downward into the ground to refusal (Douglas et al., 2016). Repeat measurements were made at flags permanently installed into the ground or using a 100m tape and high resolution gps measurements. | Douglas et al., 2016 |
| Frost | United States Tundra -- Central Yukon-Kuskokwim Delta, western Alaska | 1971, 1972, 1985, 2006, 2007, 2015 | Except in 2015 burns, thaw depths were measured at 5 m intervals along three 30 m linear transects radiating at 120° intervals from the plot centre, according to the U.S. Bureau of Land Management's Assessment, Inventory, and Monitoring Program protocol (AIM; Toevs et al 2011), providing 7 measurements per transect and 21 measurements per plot. In 2015 burns, plots consisted of four parallel 20 m transects oriented from east to west and spaced 5–10 m apart, following guidance from the Fire Effects Monitoring and Inventory System protocols; thaw depth was measured at 5 m, 10 m, and 15 m along each transect, providing 12 measurements per plot. | Frost et al., 2020 |

| | | | | |
|---|---|---|---|---|
| Gaglioti | United States Tundra -- The Noatak watershed, which drains the southwestern flank of the Brooks Range in northwestern Alaska | 1972, 1984 | Thaw depth was measured 2-3 m apart along 100-m-long transects. We used a 1.5-m-long tile probe and measured until depth of refusal. | Gaglioti et al., 2021 |
| Holloway | Canada Boreal -- Taiga Plains and Taiga Shield ecozones near Yellowknife, Canada | 2014, 2015 | Thaw depth was measured along 160 m transects with 52 measurement points per transect. At each point, a 1 cm diameter titanium probe was pushed into the ground until it met refusal. | Holloway et al., 2024 |
| Loranty | Russia Tundra -- Northeastern Siberia Larch forests | 1972 | Thaw depth measurements were taken at 1 m intervals along three 20 m transects across four burned sites within a single fire scar and four adjacent unburned. Thaw depth was quantified by measuring depth to refusal with a tile probe. | Loranty, et al., 2014 |
| Manies | United States Boreal -- Interior Alaska, black spruce forests | 1999 | Measurements within the black spruce sites occurred every 10 to 20 m along two linear transects within the site. These transects were laid out perpendicular to each other to negate any possible directional influences due to slope or dominant wind direction.. Thaw depths were measured monthly. | Harden et al., 2006, Manies et al., 2004 |
| Natali | United States Boreal & Tundra -- Bonanza Creek, Alaska USA; Anaktuvuk River fire, AK USA; Yukon Kuskokwim Delta, AK | 1983, 2003, 2004, 2007, 2015 | For Hess Creek, thaw depths were measured at 1 m intervals along 1-3 m transects that measured 20-100 m across burned and unburned sites. For Bonanza Creek, thaw depths were measured along 1-3 transects of 20-100m length every 1m. For the Anatuvuk River fire, thaw depths were measured along a transect (Natali et al., 2018). For Yukon-Kuskokwim Delta, thaw depths were measured across multiple sites across multiple years. We quantified the depth to refusal with a tile probe. | Natali et al., 2016, 2018, Natali 2018 |
| O'Donnell | United States Boreal & Tundra -- Interior Boreal, AK, USA | 1966, 1967, 1990, 2003, 2004 | For Erickson Creek fire scar, 3 replicate thaw depth measurements across ten plots per site type (upland burned, upland, unburned, lowland burned, lowland unburned) (O'Donnell et al. 2009). At Hess Creek and Taylor Highway sites, thaw depth measurements were made at 1-5 replicate plots per stand age (O'Donnell et al. 2011a, 2011b, 2013). We quantified the depth to refusal with a tile probe. | O'Donnell et al., 2009, 2011a, 2011b, 2013 |
| Olefeldt | Canada Boreal -- Western Boreal Canada | 1964, 1967, 1975, 1982, 1984, 1995, 2000, 2006, 2007, 2008, 2012, 2013, 2014, 2019 | At each site we collected 100 thaw measurements in a 30 x 30 m grid, with measurement points every 3 m. We quantified the depth to refusal with a 150 cm steel probe. | Gibson et al., 2018 |
| Paulson, Alexander | Russia Boreal -- | 1983, 1984, 1990, 2001, 2002, 2003, 2010, 2015 | Within each plot, we measured thaw depth 5 times along a 20 m S - N transect, at 0, 5, 10, 15, and 20m within each plot along 1-3 transects across 13 fire scars. We quantified the depth to refusal with a tile probe. | Alexander et al., 2020 |

| | | | | |
|---|---|---|---|---|
| | Northeastern Siberia near Cherskiy, Russia, and Yakutsk, Russia | | | |
| Rocha | United States Tundra -- North Slope of Alaska | 1977, 1993, 2001, 2007 | CALM grid plus transects at 1-12 year old sites (Rocha and Shaver 2011), and transects only at other sites. We quantified the depth to refusal with a tile probe | Rocha and Shaver, 2011 |
| Sizov | Russia Tundra -- Northwestern Russia, Nadym region of the Yamal-Nenets Autonomous Okrug | 2020 | Across seven sites, temperature was measured in shallow boreholes with a Tr 46908 thermometer (TR di Turoni & c. Snc, Italy) and drilling was carried out using a hand-held motor-drill Stihl BT 360 (Stihl, Germany). Measurements occurred in mid-August, at approximately 10cm increments. | Sizov et al., 2020 |

179

180

181

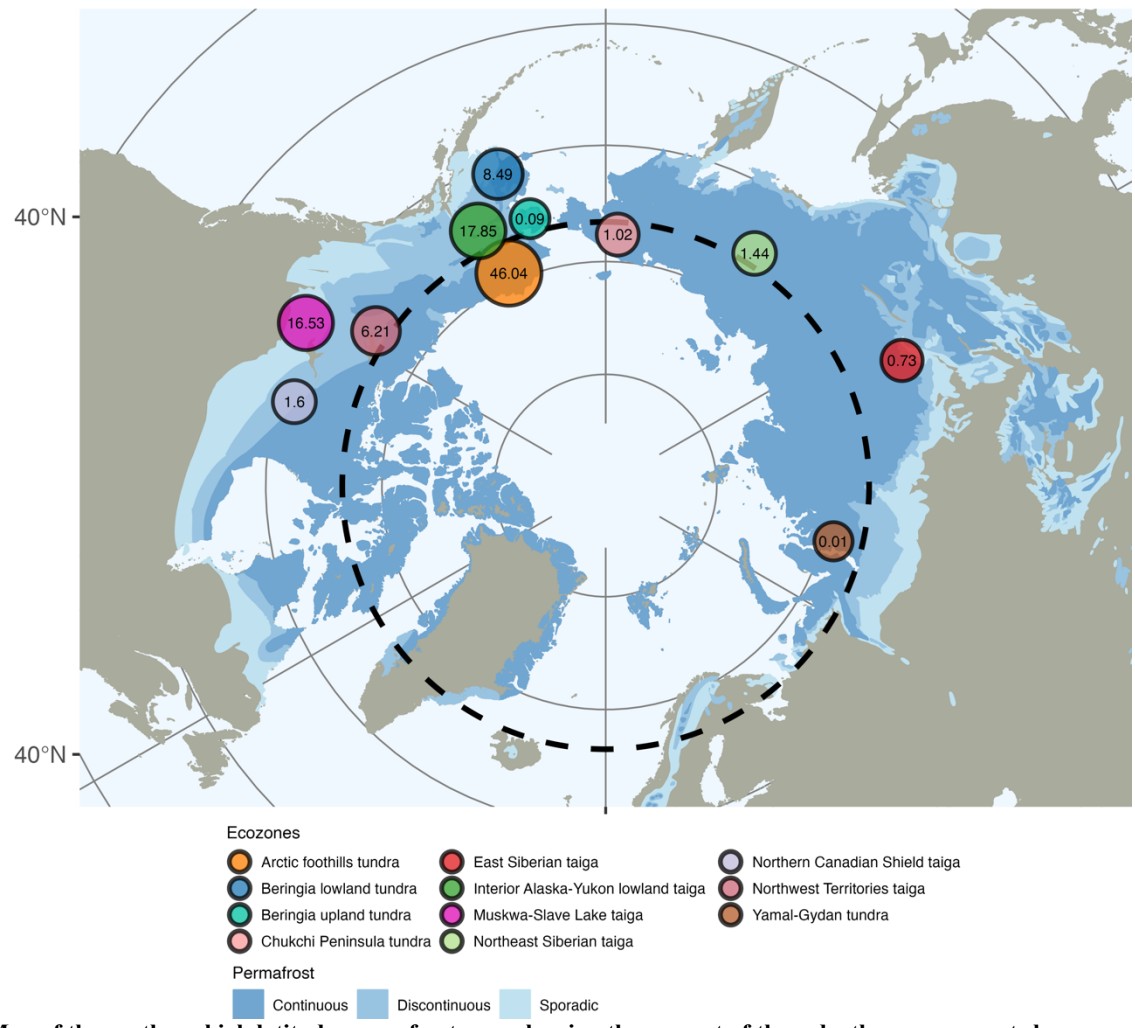

**Figure 1. Map of the northern high latitude permafrost zone showing the percent of thaw depth measurements by ecozones (circle colour, Dinerstein et al., 2017) with the extent of continuous, discontinuous, and sporadic permafrost shown in shades of blue (Brown et al., 1998). Points are sized and labelled with the percent of measurements within each ecozone. The Arctic circle is shown with the thick dashed black line.**

We screened the data for issues with units, sign convention, coordinates, and data type (e.g., factor, integer). Where we required categorical variables, we ensured these were spelled in a consistent manner and that the correct unique number of variables were returned. We mapped the data to check inaccurate site coordinates and checked discrepancies, such as missing negative signs from longitude, with contributors. We used histograms of measurement depths to identify any outliers in the data, several

of which were removed after confirming with the contributors that they were the result of typographic errors. Data contributors
were asked to note if any measurements hit rock, and, when noted, these observations were excluded from the final dataset.

**2.2 Estimating Active Layer Thickness**

Over the course of the growing season, the depth of the thawing front increases as the active layer expands to its maximum.
Therefore, measurements taken throughout the thaw season are not directly comparable with one another. Therefore, we
standardised thaw depths taken at different times throughout the thawing season, which resulted in an estimated dataset of
ALT. To do so, we estimated ALT using a modified version of the Stefan equation, used by Holloway and Lewkowicz (2020)
and described by Riseborough et al. (2018) and Bonnaventure and Lamoureux (2013). Estimating ALT (Fig. 2) allows thaw
depth measurements collected during different times in the growing season to be comparable and used to understand the full
effects of wildfire on the active layer across paired sites in a given measurement year and for some of the sites across multiple
203 years.

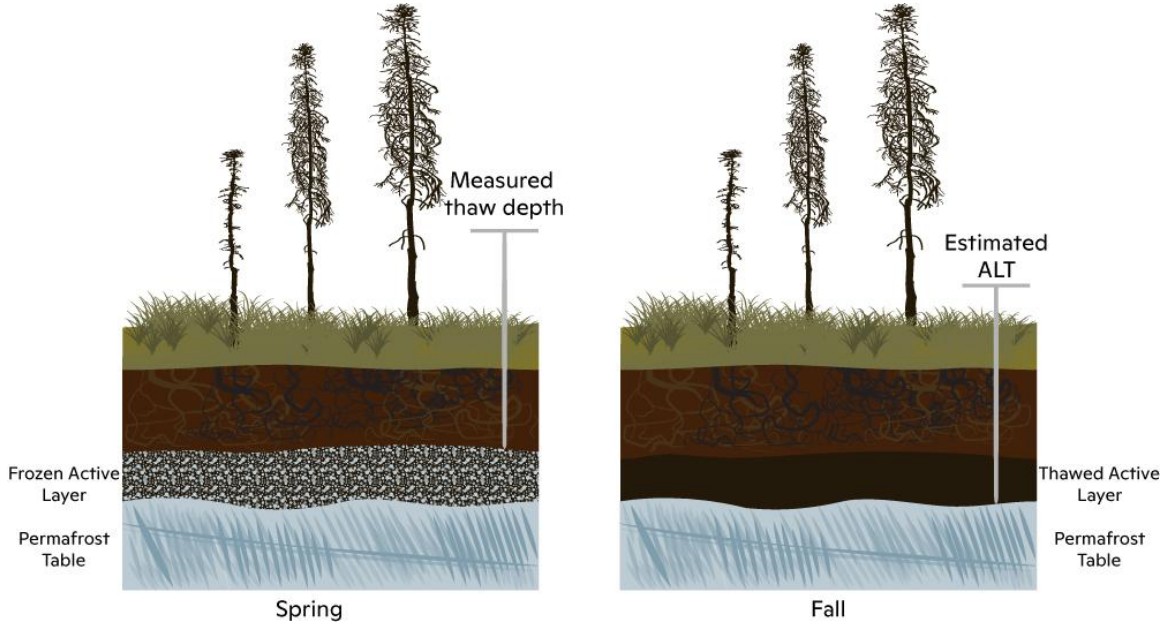

**Figure 2. Diagram of early season thaw depth measurement versus late season active layer thickness. The active layer expands**
**during the thawing season reaching its maximum thickness between August and November depending on the location.**

ALT was estimated based on air thawing degree days (TDD; i.e., days above zero degrees Celsius during the thawing season).
Others have shown a correlation between TDD and ALT (e.g., Strand et al., 2021). Daily mean air temperatures were extracted
from ERA5-Land daily aggregates (Muñoz Sabater 2019) accessed through Google Earth Engine (Gorelick et al., 2017).
Instrumental air temperature data are sparse across the northern high-latitude regions. We selected the ERA5-Land (Muñoz
Sabater, 2019) dataset since it is available for the full region and time series, accessible through Google Earth Engine, and has

been evaluated against meteorological station data (Rantanen et al., 2023, Clelland et al. 2024). Across the circum-Arctic and Asian boreal ERA5-Land validation studies indicate a warming bias in winter months of a half a degree Celsius (Rantanen et al., 2023, Clelland et al. 2024), whereas validation studies in summer indicate a slight cooling trend of ~0.2 degrees Celsius (Rantanen et al., 2023). Due to the scarcity of meteorological stations across the Northwestern Territories, we provide additional validation for air temperature data from ERA5-Land using shielded air temperatures at a height of 1.5 m that were measured at six sites using Onset Corporation (USA) Hobo Pro U23-003 loggers (accuracy ±0.21°C; precision ±0.02°C). All air temperature data were aggregated from 2-hour samples to daily averages and sites included thaw depth measurements (Holloway 2020). We calculate Pearson's correlation coefficient ($R$), bias (defined as the summation of modelled minus measured divided by the number of data points), and the root mean square error (RMSE). The correlation is ~0.99, with a warming bias of 0.54 degrees Celsius, and a RMSE of 2.23 degrees Celsius (Fig. S2).

First, we defined the end of the thaw season for each measurement location and year based on when the five-day mean daily air temperature shifted from above- to below-freezing. We then subtracted 14 days from the end-of-season date to account for the lag between surface freezing and the refreezing of the bottom of the active layer. Typically, the active layer begins to freeze upward while the air temperature is still above zero, requiring approximately 7-14 days until the surface freezes (Osterkamp and Burn 2002). Following the Stefan equation (Freitag and McFadden, 1997), we calculate (A) as the square root of the sum of daily mean air temperature TDD prior to the day of year of the field measurement (i.e., thaw depth), as in Eq. (1):

$$A = \sqrt{\sum_{TDD\ thaw\ depth=1}^{n} TDD\ Thaw\ depth}\ , \tag{1}$$

We calculate (B) as the square root of the sum of daily mean air temperature TDD (i.e., days above zero degrees Celsius) prior to the end of thaw season day of year (i.e., ALT) Eq. (2):

$$B = \sqrt{\sum_{TDD\ ALT=1}^{n} TDD\ ALT}\ , \tag{2}$$

Finally, we multiplied the field measured depth by the ratio of the first two equations to calculate the estimated ALT Eq. (3):

$$estimated\ ALT\ =\ field\ measured\ depth\ \times\ (B \div A)\ , \tag{3}$$

An example of the calculation for two sites is provided in Table 3 and shown in Fig. 3.

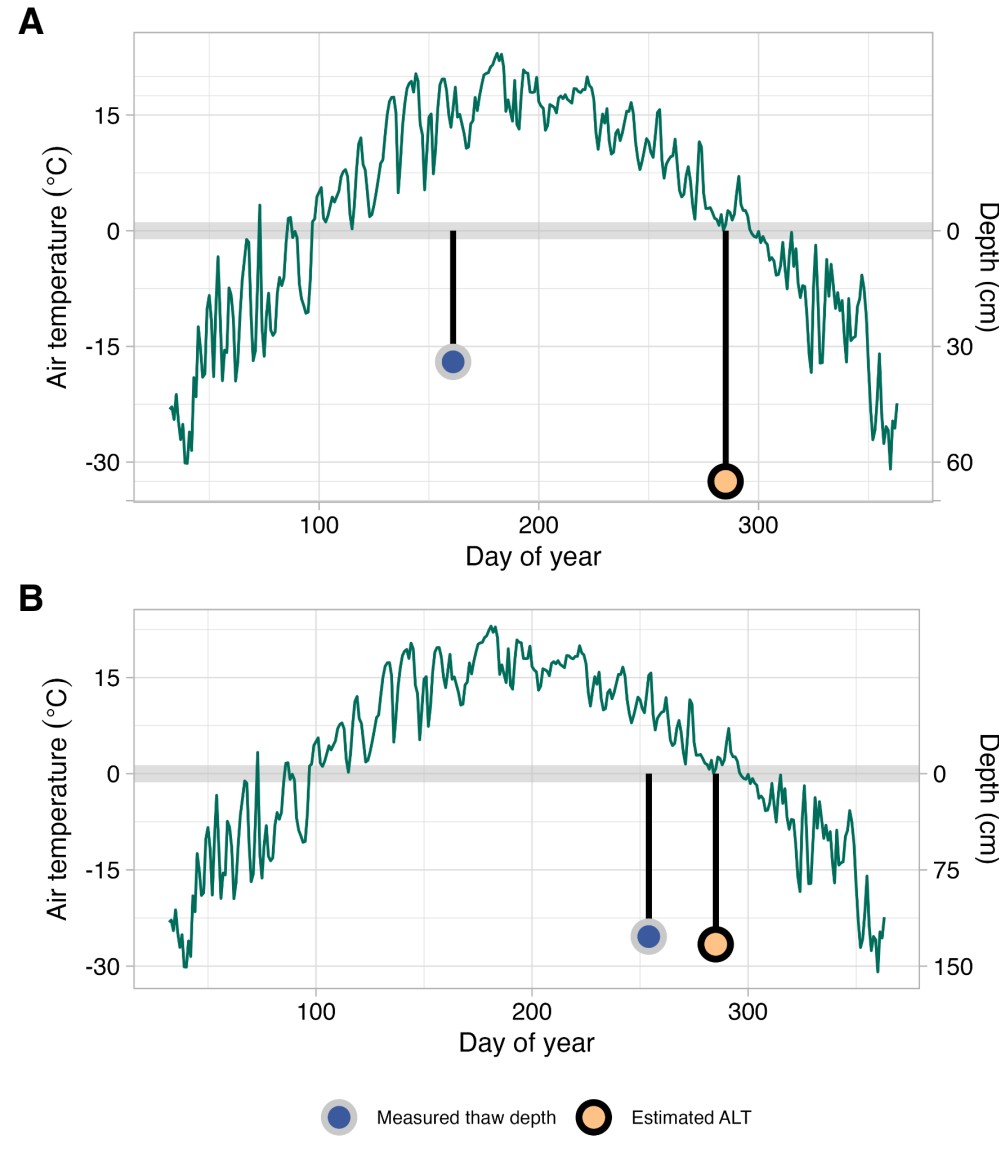

Measured thaw depth     Estimated ALT

**Figure 3. An example of estimating active layer thickness from two *in situ* thaw depth measurements using seasonal air temperature.**
**Air temperature through the thawing season (green line) for two separate sites, one with an early-season thaw depth measurement**
**(A) and a second with an end-of-season thaw depth measurement (B). For each site, we show the measured thaw depth (blue point)**
**and estimated ALT depth (orange point) for the day of year either measured or estimated. The right y-axis shows thaw depth (cm),**
**the left y-axis shows air temperature and the x-axis shows the day of the year.**

**Table 3. An example of estimating ALT using Equations 1-3 from two *in situ* thaw depth measurements at two sites (A and B) using**
**the same data as in Fig. 3.**

|  | Site | A | B |
|---|---|---|---|
| **Data contribution** | Timing of measurement | Early season | End of Season |

|  | Year | 2015 | 2015 |
|---|---|---|---|
|  | Month | 6 | 9 |
|  | Day | 10 | 11 |
|  | Day of year | 161 | 254 |
|  | Measurement depth (cm) | 34 | 127 |
| **Calculated from ERA5 data extracted based on location** | Day of year first of five consecutive days at zero | 299 | 299 |
|  | Day of year to estimate ALT | 285 | 285 |
|  | Eq.1 | 25.25 | 45.95 |
|  | Eq.2 | 48.03 | 48.03 |
| **Estimated ALT** | Eq.3 (cm) | 65 | 133 |

Estimates were excluded for observations that hit rock, were greater than the depth of the measurement probe, or were missing

the day of month (Table S2). We were unable to convert every early season thaw depth to ALT if the date of measurement

was not preceded by at least one day above zero degrees Celsius, in which case these measurements were removed from the

estimated dataset. Ultimately, 48,669 of the original 52,466 measurements were included in the estimated dataset.

**2.3 Quantify uncertainty of estimated ALT**

We quantify uncertainty in our estimates of ALT by calculating Pearson's correlation coefficient ($R$), bias (defined as the

summation of modelled minus measured divided by the number of data points), and the Root Mean Square Error (RMSE). The

bias indicates whether estimated ALT is over or underestimated, while the RMSE provides an average error regardless of sign.

We used two data sets for this analysis from contributors that had repeat measurements from within a season for early/mid-

season and late season at the same locations.  These data sets differed as one was a subset of their data contributed to the data

synthesis for the boreal near Yellowknife, Canada (N = 626; Holloway et al. 2024), whereas the other was used solely for

quantifying uncertainty for tundra on the Seward Peninsula, AK (N = 37; Breen, unpublished). The tundra data was missing

key meta data which precluded it from the synthesis.We used the early/mid-season measurements to estimate thaw depths for

the date of the late season measurement (as opposed to the end of the thaw season defined using ERA5-Land) following the

methodology described in Section 2.2, to quantify the uncertainty in the estimation process.

**2.4 Spatial attributes**

We added spatial attributes to the data through spatial joins. We generated a point shapefile using the latitude and longitude

coordinates with the coordinate reference system (CRS) 4326 (i.e., WGS 84). We performed a spatial join to add ecozone data

(Dinerstein et al., 2017), retaining the ecozone and biome names. We then performed a second spatial join with permafrost

data (Brown et al., 1998), retaining permafrost extent (e.g., continuous, discontinuous, sporadic). We show the distribution of

estimated ALT measurements by ecozone (Fig. 4). The spatial coverage, and hence inherent resolution, of these polygon

products is much larger than the data points or any site-level aggregation. Due to the coarser resolution, data contributors'

designation of biome outweighed what was assigned through the spatial join. The small percentage of plots where the biome

was misassigned were visually inspected and found to be adjacent to the boundary with the matching biome and were manually
reassigned (see code).

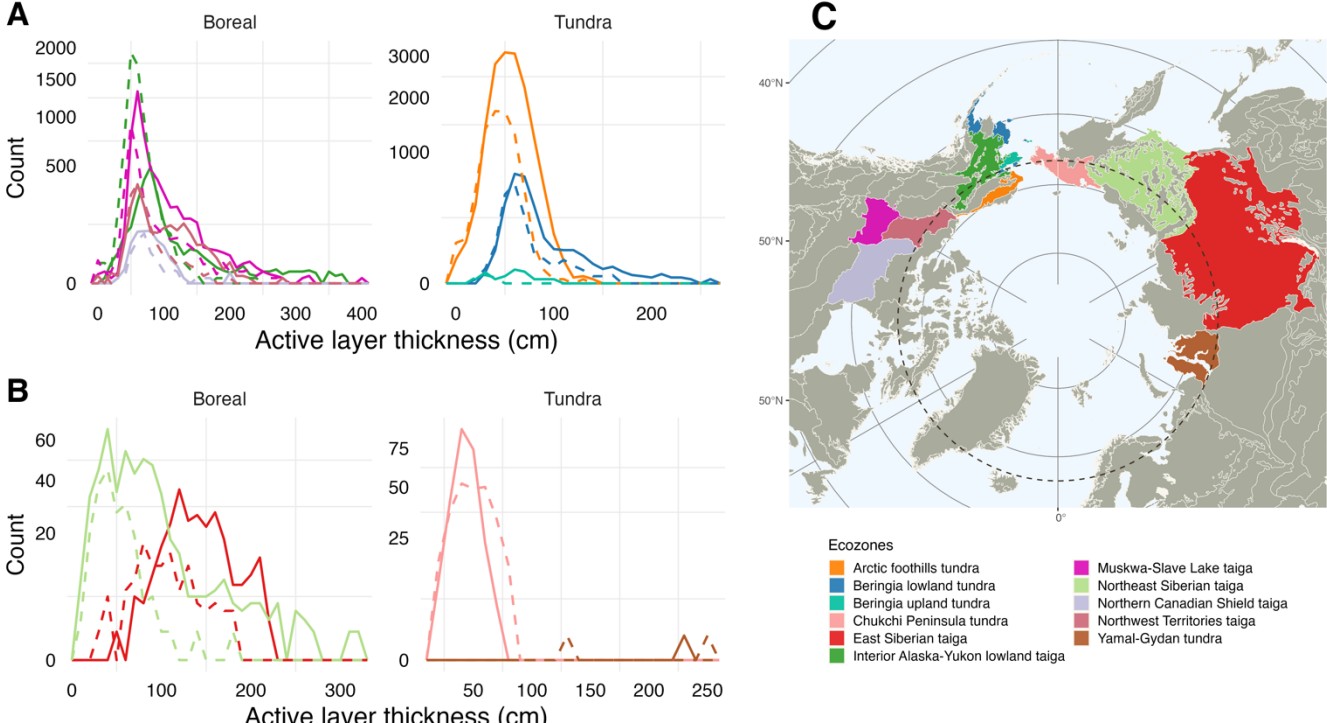

**Figure 4. Frequency distribution graphs showing estimated active layer thickness (cm) by ecozones split by North America (A) and**
**Eurasia (B), solid line for burned distribution and dashed line for unburned distribution. Map of ecozones for location reference (C;**
**Dinerstein et al., 2017). The y-axis is the count of measurements and the x-axis is the depth in centimetres. Both x- and y-axis vary**
**by panel and y-axes are adjusted to show low counts.**

## 2.5 Data structure and columns

The resulting dataset includes 32 attributes including attributes from the initial contribution, plus the attributes from the spatial
joins and the derived ALT estimates all described in Table 4. The dataset is shared in comma separated values (csv) format
with 48,669 rows and 32 columns. For missing values, we used 'NA' and '-9999', for character and numeric fields,
respectively.
**Table 4. Description of data attributes and data format. All attributes are included with the raw data. Attributes included with the**
**plot level data are denoted with a * and data from paired burned/unburned are denoted with a †.**

| Attribute | Format | Description |
| --- | --- | --- |
| plotId* | character | A unique identifier assigned by the data contributor to identify the field plot. |
| siteId* | character | Site name assigned by the data contributor specific to the fieldwork. |

| | | |
|---|---|---|
| lastNm*[†] | character | Last name(s) of the person(s) contributing the data provided by the data contributor. |
| submitNm*[†] | character | Last name of the data contributor that submitted the form (single name only). |
| biome*[†] | character | Boreal (B) or tundra (T) assigned by the data contributor. |
| distur*[†] | character | Categorical variable to identify location as burned or unburned provided by the data contributor. |
| cntryId*[†] | character | Dropdown list of two-digit code: Russia (RU), USA (US), Canada (CA), Finland (FI), Norway (NO), Sweden (SE), Iceland (IS), Greenland (GL) assigned by the data contributor. |
| fireYr*[†] | integer | Four-digit year of when the fire event occurred provided by the data contributor. |
| fireId*[†] | character | Unique fire identifier assigned by the data contributor. |
| gtProbe* | character | Permafrost thaw depth exceeds (i.e., greater than [gt]) the length of probe yes (y) or no (n) provided by the data contributor. |
| hitRock* | character | Probe hit rock yes (y) or no (n) provided by the data contributor. |
| lat* | float | Latitude in decimal degrees in WGS 84 provided by the data contributor. |
| lon* | float | Longitude in decimal degrees in WGS 84 provided by the data contributor. |
| year*[†] | integer | Four-digit year the data were collected provided by the data contributor. |
| month | integer | Two-digit month (values 01-12 accepted) the data were collected provided by the data contributor. |
| day | integer | Day of month data were collected values( 1-31) provided by the data contributor. |
| orgDpth* | integer | Organic layer thickness measured from the ground/moss surface to the organic-mineral interface, as a site mean in cm, provided by the data contributor. |
| srfH2O* | character | A categorical variable describing if plot locations experience seasonal inundation (i.e., standing surface water during the early season but dry by late season). Seasonal inundation (Y: yes) or not (N: no) or unknown (U). Provided by the data contributor. |
| msrType | character | A categorical variable of thaw (T) or active (A). Active refers to active layer thickness (i.e., maximum seasonal thaw at the end of growing season), and thaw refers to thaw depth (i.e., less than seasonal maximum taken earlier than the end of thawing season). Provided by the data contributor. |
| msrDoy | integer | Day of year (DOY) for the day of measurement converted from YYYY-MM-DD. |
| msrDepth | float | The field measurement of the thaw depth or ALT in cm. Provided by the data contributor. |
| topoPos* | character | Categorical variable describing the topographic position of plot locations as upland (U), midslope (M), lowland (L). Provided by the data contributor. |
| slope* | integer | Numeric value indicating slope angle provided by the data contributor. |
| vegCvr* | character | Evergreen needle-leaf (EN); broadleaf deciduous (BD); deciduous needle-leaf (DN); mixed needle-leaf majority MNM; mixed (M); mixed broadleaf majority (MBM); barrens (B), graminoid tussock dominated (GT), graminoid non-tussock dominated (GNT), prostrate shrub dominated (P), erect-shrub dominated (S), and wetlands (W). Provided by the data contributor. |
| resBiome* | character | Biome assigned by spatial join with the Resolve data product (vector data) 'BIOME_NAME' (Dinerstein et al., 2017). |
| resName* | character | Ecozone name assigned by spatial join with the Resolve data product (vector data) 'ECO_NAME' (Dinerstein et al., 2017). |
| permaExtent*[†] | character | Permafrost extent (vector data) assigned by spatial join with permafrost ground-ice map 'EXTENT' as C=continuous, D=discontinuous, S=sporadic (Brown et al., 1998). |
| estDoy* | integer | The day of year used to estimate ALT based on when the five-day mean daily air temperature shifted from above- to below-freezing. |
| estDepth*[†] | float | The estimated ALT in cm; calculated using air temperature from ERA5-Land and field measured thaw depth. |

| paired*[†] | character | Identifying code to pair unburned measurements to burned measurements provided by the data contributor. |
|---|---|---|
| tsf*[†] | integer | Time since fire calculated by subtracting year from fireYr. |
| tsfClass*[†] | character | Binned time since fire (tsf) classes in years as "unburned", "0-3", "4-10", "11-20", "21-40", ">40" |
| n*[†] | integer | number of measurements used to calculate plot-level or pair burned/unburned means |

**2.6 Aggregating plot level means and burned to unburned pairs**

While the main objective of the data synthesis is to provide paired burned/unburned ALT estimates, we also want to provide details on aggregating to the site/plot level. We aggregated plot and paired level data in R with 'tidyverse' (Wickham et al., 2019). Plot level data was aggregated using the `group_by` function aggregate using the following variables: data contributor ('submitNm'), burned or unburned ('distur'), site level identifier ('siteId'), plot level identifier ('plotId'), fire year ('fireYr'), and year of measurement ('year'), which captures both the spatial and temporal component of the data. We then calculated the mean ALT for each plot that includes 28 attributes (see Table 4 for descriptions). Paired burned and unburned sites are a unique and defining characteristic of this dataset. Data contributors were required to provide details on how their burned measurements paired with unburned measurements. Characteristics of unburned plots were required to be representative of biogeoclimatic conditions prefire and within close proximity to their paired burned plot(s). The dataset includes a code to link burned with unburned sites ('paired'). To aggregate at the paired level, we grouped by data contributor ('submitNm'), burned or unburned ('distur'), pairing code ('paired'), year of the fire event ('fireYr'), and can be further grouped by time since fire ('tsf'). The paired burned/unburned data includes 13 attributes (Table 4).

**3 Data summary**

**3.1 General Characteristics of the data**

In total, the final dataset includes 48,669 observations from the original 52,466 observations across 9,446 plots and 388 sites. Thaw depth measurements are predominantly from North America, with 35,272 (19,739 burned, 15,533 unburned) in Alaska and 11,844 (7,553 burned, 4,291 unburned) in Canada, and 1,553 (998 burned, 555 unburned) in Russia. These *in situ* measurements were collected within the continuous, discontinuous, and sporadic permafrost zones (Fig. 1). Data were contributed with both burned and unburned paired sites with fire years ranging from 1900 to 2022 across 112 fire events. There are 193 unique paired burned/unburned measures based on pair id (76), fire year (37 unique years), fire events (63 unique events), and time since fire spread across 11 ecozones. There are 21,589 estimated observations across the boreal forests/taiga

and 27,080 estimated observations across the tundra biomes (Fig. 4). There are 27,638 observations from continuous permafrost, 12,905 from discontinuous permafrost, and 8,126 from sporadic permafrost.

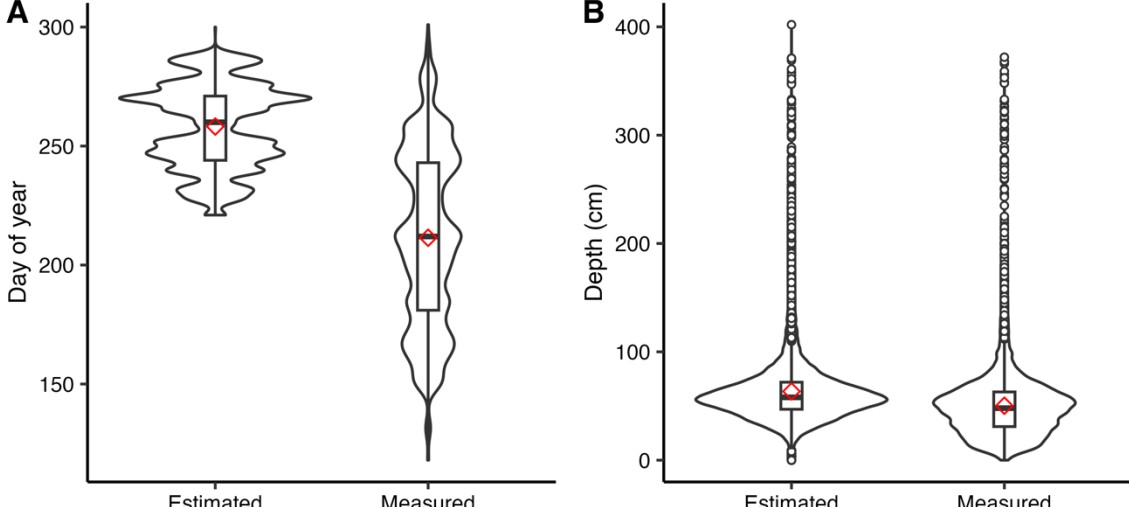

**Figure 5. The distribution for *in situ* measurements vs. estimated measurements. For day of year (A) and thaw depth (B), we show the distribution for *in situ* measurements vs. estimated measurements using violin plots overlain with boxplots with a red diamond marking the mean. Measured day of year and depths were provided in the raw data contribution. The day of year shows a wide spread of dates, which is caused by the broad geographic extent of the data. Estimated values were calculated to create a dataset that characterises maximum thaw depth (i.e., ALT).**

**3.2 Estimated ALT**

The estimated ALT provides a temporally consistent measurement capable of quantifying the effects of wildfire on active layer dynamics temporally and spatially. The data show the shift from measured thaw depth to estimated ALT characterised by a narrower range of dates and depth measurements (Fig. 5A & 5B). The day of year is condensed for the estimated measures (Fig. 5A), which was anticipated since the contributed data were collected throughout the thawing season resulting in a wide spread due to the broad geographic extent of the data whereas the estimated data were truncated to the later part of the thaw season, resulting in a narrow range of days. The uncertainty in the estimated ALT varies with biome and disturbance (Table 5, Fig. 6). Boreal burned values tend to underestimate by about five percent, whereas unburned values tend to overestimate by about 15 percent. For the tundra, burned and unburned values tend to be overestimated by 19.6 and 22.8 percent respectively. The sample size is much smaller for the tundra biome for estimating uncertainty.

**Table 5. Quantifying uncertainty for estimated ALT. We report the root mean square error (RMSE), percent uncertainty, mean**
**residual error as an indication of bias, and sample size for burned and unburned sites in the validation dataset. Negative values**
**indicate an overestimation and positive values indicate an underestimation.**

| Biome | Disturbance | RMSE | Percent uncertainty | Mean residual error (bias) | Sample size |
|---|---|---|---|---|---|
| **Boreal** | **Burned** | 22.8 | 4.6 | 5.7 | 413 |
| **Boreal** | **Unburned** | 20.3 | 14.5 | -8.4 | 212 |
| **Tundra** | **Burned** | 29.2 | 19.6 | 13.9 | 20 |
| **Tundra** | **Unburned** | 5.6 | 22.8 | 12.5 | 6 |

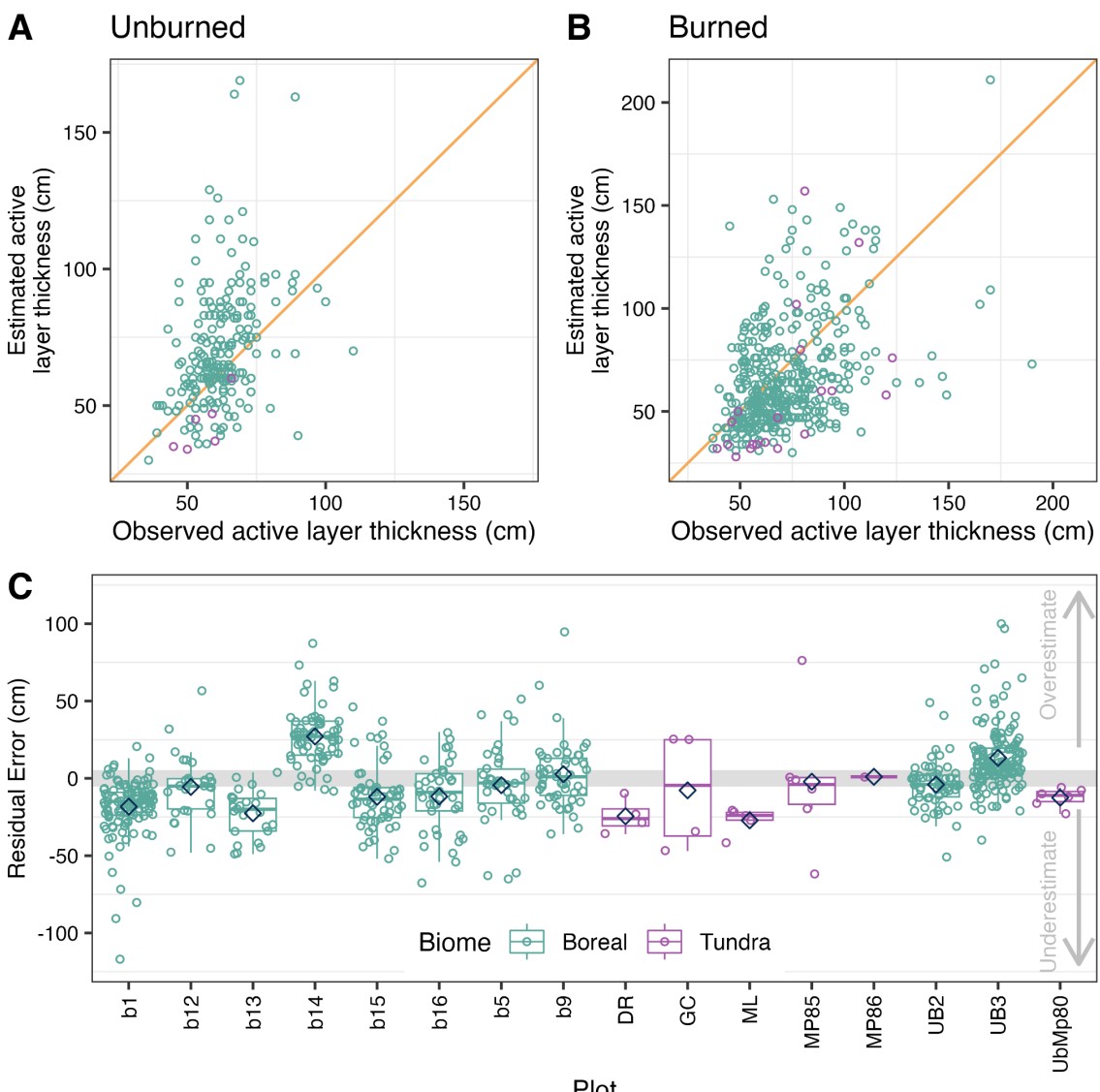

Figure 6. Quantifying uncertainty of ALT estimates. Panel (A) and (B) show observed depths compared to estimated depths split by
unburned and burned plots with the orange line showing a slope of one. Panel (C) shows the bias by plot identifier, where zero
indicates no difference between the observed and estimated values. Negative values indicate an underestimation and positive values
indicate an overestimation with the mean shown by the blue diamond. Burned sites include b1, b12, b13, b14, b15, b16, b5, b9, DR,
GC, ML, MP85, and MP86, and unburned sites are ub2, ub3, and UbMp80.

### 3.3 Difference in estimated ALT between burned and unburned sites

By aggregating the burned and unburned pairings, we show the percent difference in estimated ALT between burned and unburned sites post-fire (Fig. 7, S3). Most sites show a thickening of the active layer post-fire compared to adjacent unburned sites. Generally, across boreal sites the mean percent difference shows a thickening of the active layer in the two decades following fire, followed by a recovery in the subsequent decades (e.g., time since fire 21-40 and >40). The magnitude of difference varies by biome and permafrost extent. In the boreal forest continuous permafrost region, the means follow this general trend of expansion followed by recovery, however, there is very limited and no data at 4-10 years and >40 years, respectively. The boreal forest discontinuous permafrost region follows the general trend, whereas the boreal forest sporadic permafrost region shows a lower percent difference in the two decades following fire where the active layer does expand but not to the same extent as seen in the continuous or discontinuous permafrost following a varied recovery at 21-40 and >40 years. The tundra biome follows the same general trend that the boreal sites do where mean percent difference shows a thickening of the active layer in the two decades following fire, followed by a recovery in the subsequent decades (e.g., time since fire 21-40 and >40). This trend is most distinct for tundra sites with continuous permafrost, whereas sites with discontinuous permafrost show a bit more variability for 11-20, 21-40, and >40 years. The tundra sites with discontinuous permafrost have a sample of one for 21-40 and >40 years, which makes it challenging to fully understand the recovery trend. The trend of post-fire thickening of the active layer followed by recovery illustrates the effect of climate on permafrost recovery. The variability in the extent of the thickening of the active layer across permafrost zones might provide insight to potential future patterns. Specifically, the reduced thickening seen in the warmer boreal sporadic region might be a future pattern that we see extending to the boreal discontinuous zone as the climate continues to warm.

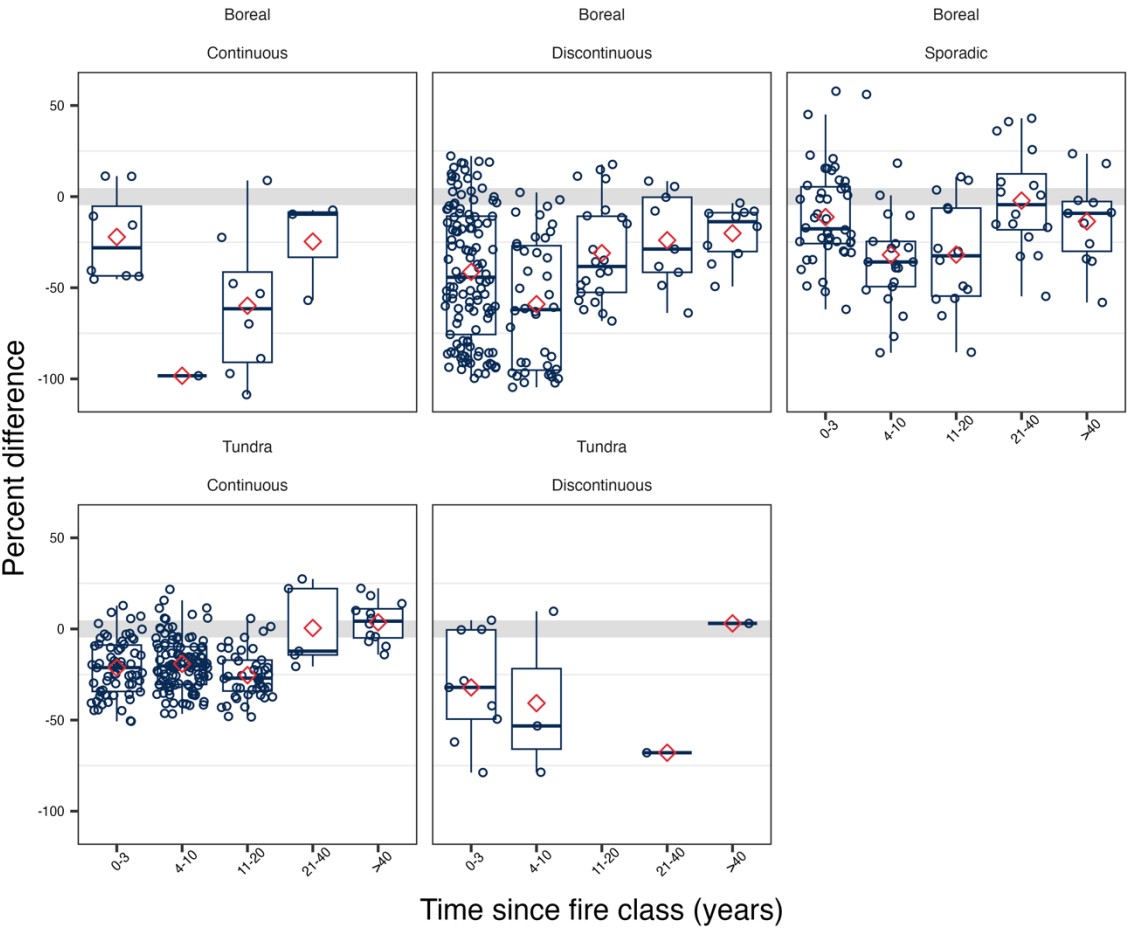

**Figure 7. Percent difference in estimated ALT between burned and unburned paired sites in the years following wildfire. The percent difference is calculated (unburned-burned)/((unburned + burned)/2) * 100. Negative values indicate that the burned sites have a thicker active layer than the unburned site, while values around zero show little difference in ALT, and positive values indicate that unburned sites have a thicker active layer than the burned ALT. The red diamond indicates the mean based on paired burned-unburned and then aggregated by time since fire class, permafrost extent, and biome. The box and whisker plots show the split in quantiles. See Supplemental Materials to see a similar plot by ecozone (Fig. S3).**

## 4 Strengths, Limitations, and Opportunities

### 4.1 Strengths

The FireALT dataset (Talucci et al., 2024) offers paired burned and unburned sites that can be aggregated and viewed both spatially and temporally to provide critical insights for understanding wildfire impacts on ALT, a feature commonly used to determine permafrost conditions. Field data collection is often spatially and temporally opportunistic, making comparisons of disparate datasets difficult. For example, several geographically similar sites had depth measurements collected across a wide

range of dates throughout August and September, but these measurements were not necessarily capturing the maximum ALT and therefore not comparable. Further, it is challenging to compare early to end of season thaw depth measurements (Holloway and Lewkowicz 2020). By estimating ALT, the data can be used to extrapolate beyond individual measurements and provide broader understanding of spatial and temporal feedbacks between wildfires, permafrost, and climate. Additionally, data include several environment attributes, e.g., organic layer depth, slope, topographic position, and whether surface water was present. Future analyses could integrate these environmental variables to expound upon the relationship between environmental variables, ALT, and wildfire. Finally, we show a general expansion of the active layer following fire followed by recovery 40 years post-fire but the magnitude of expansion and recovery vary by biome and permafrost zone, pointing to the role of vegetation, permafrost conditions, and climate on active layer dynamics in response to wildfire (Brown et al., 2015). Climate has changed over the time period of the fire events captured within this dataset. Generally, the data indicates that we may expect the active layer to fully recover 40 years post-fire, but that may change for more recent fires. The boreal sporadic zone experiences less expansion of the active layer with a less distinct recovery, which demonstrates how climate influences active layer recovery in warmer regions. This illustrates how climate influences permafrost recovery, and with a warming climate, we may expect to see patterns more like this in boreal discontinuous permafrost zones.

## 4.2 Limitations, uncertainty, and bias

Estimating ALT is crucial for spatial-temporal evaluations of wildfire-permafrost interactions due to the variability in thaw depth throughout the thaw season. However, uncertainties arise in the estimated ALT from the data we integrate to make those calculations. Air temperature can be a reliable metric for calculating maximum ALT (Osterkamp and Burn 2002, Holloway and Lewkowicz 2020), but the coarse resolution climate data and *in situ* weather station gaps (Clelland et al. 2024), as well as the lack of accounting for disturbance effects on air temperature (Kurylyk and Hayashi, 2016, Muñoz-Sabater et al., 2021, Helbig et al., 2024), all impact the accuracy of the estimated ALT. The Stefan equation assumes negligible soil heat capacity and thus can overestimate thaw depth, and it also does not account for fire altering the surface energy balance (e.g., reducing albedo, loss of canopy and shading) and heat fluxes (e.g., loss of above-ground biomass), all of which increase thaw depths and can contribute to  underestimations of ALT (Kurylyk and Hayashi, 2016). Our quantification of uncertainty supports this underestimation bias for burned sites and over estimation for unburned sites in the boreal biome. Further, the lack of inclusion of frozen water content in the Stefan equation may affect early season measurements due to the zero curtain, where the rate of thawing may not scale directly with air temperature (Osterkamp, 1987, Romanovsky and Osterkamp, 2000). These effects likely vary between tundra and boreal sites. These are dynamic systems with multiple feedbacks that influence the freeze-thaw cycle and the timing of maximum thaw depth. Similarly, the time at which permafrost begins to refreeze from the bottom varies with permafrost temperature, soil moisture and thermal properties, and local edaphic hydrological conditions. Consequently, our assumption that ALT occurs 14 days before the date at which air temperature drops below freezing is

another source of uncertainty. Overall, interannual variability in ALT is dependent on complex interactions between air temperature, precipitation, snow dynamics, hydrothermal processes, water energy exchanges, and fluctuations in thaw season length, which are a source of uncertainty in our approach (Shur et al., 2005, Hu et al., 2023, Grünberg et al., 2024). While in warmer boreal sites the 14 day lag may be longer or non-existent depending on the complex interactions of these landscape-level controls. Despite this, estimating ALT allows for insightful comparisons between sites that are not appropriate or meaningful with the raw data.

Burn severity is a critical component of wildfire that impacts ALT and permafrost stability through combustion of the insulating organic matter, vegetation and post-fire changes in albedo (Rocha and Shaver 2011, Alexander et al., 2018). We do not account for burn severity in the data, which could strongly influence differences we see between burned and unburned ALT. Burn severity could be estimated using the organic depth measurement in the data, but the organic depth will be influenced by time since fire or through the integration of satellite imagery that could be used as a proxy for burn severity. However, vegetation indices that estimate burn severity (e.g., differenced Normalized Burn Ratio [dNBR]) are typically better correlated with aboveground burn severity while less indicative of burn depth (e.g., Delcourt et al., 2021). Recent research which has shown combinations of remote sensing proxies, dNBR, and land surface temperature could be used in conjunction with these field measurements to estimate changes in ALT across fire scars (Diaz et al., 2024). Additionally, the ice content of permafrost may impact the interaction between wildfire and permafrost, with direct effects on ALT particularly where subsidence is involved or where the increase in ALT contributes to the degradation of ice-rich permafrost (e.g., Yedoma) in the short-term (Nelson et al., 2021, Strauss et al., 2021, Jones et al., 2024). Subsidence is not accounted for in the synthesised data. Subsidence can introduce additional bias in the measurement of ALT since thaw depth probing uses the surface as a reference. In areas where subsidence had occurred after fire, our data set will underestimate the magnitude of active layer thickening caused by fire. Bias from subsidence is difficult to estimate because it would be spatially heterogeneous, temporarily nonlinear, and largely dependent on ice content (Shiklomanov et al., 2010, O'Neill et al., 2023, Painter et al., 2023).

In addition to these physical controls, there are additional biogeomorphic factors that influence changes in ALT from fire. Landscape scale variation in topography, soil type and moisture, ground ice content, and vegetation cover and regrowth are all sources of uncertainty that cannot be accounted for in our synthesised dataset (Shiklomanov et al., 2010, O'Neill et al., 2023, Painter et al., 2023) accounting for these drivers would require datasets that may or may not be available, and is a separate research effort outside the scope of this paper. We use ecozones to highlight summary statistics of the data set since ecozones are characterised by sharing similar climates, geologic substrates, vegetation, and landforms. The use of ecozones for providing a broad overview of the data, which captures some of the variability in ALT measurements; however, finer-scale landscape features likely still add substantial variation to the estimated ALT and changes from fire. Future work could analyse how microtopographic features that influence local hydrology, burn severity, vegetation structure and function, and ice content impact wildfire-induced changes in ALT. Further, while growing season lengths and thawing degree days have increased over

the last century (e.g., Barichivich et al., 2012), the data synthesised here was only measured from 2001 onward despite covering fire events from 1900-2022. Recent thaw depth measurements from areas that burned more than several decades ago represent a post-fire evolution of the active layer under climatic conditions that no longer exist. The snapshot of thaw depth related to wildfire events in space and time provided by this data set may therefore include climatic effects that are hard to disentangle under current warming trends (e.g., Liu et al., 2024), which may bias the estimated ALT.

## 4.3 Representativeness of the data

The data included in our dataset are predominantly from North America, and there are large spatial gaps across the northern high latitude permafrost region (Fig. S4). For example, Russia is underrepresented despite containing 65% of the northern high-latitude permafrost (Anisimov and Reneva 2006, Streletskiy et al., 2019) and a majority of the burned area within the northern permafrost region (Loranty et al., 2016). The lack of data for this region is further exacerbated by the Russian invasion of Ukraine (López-Blanco et al., 2024), which has impacted international collaborations. Additionally, some of the spatial gaps could be a function of the submission criteria that required a burned/unburned pair. Due to the remoteness of northern high latitude fires, field campaigns may be constrained spatially and temporally based on accessibility of field sites and timing of field campaigns. Opportunistic site selection introduces bias into the dataset; however, this is unavoidable for the data synthesis effort that relies on contributions of existing data.

## 4.4 Future research opportunities

There is opportunity to expand this dataset to increase the spatio-temporal coverage of the data to better understand impacts of wildfire on permafrost dynamics. While we touch on how ALT differs across burned and unburned sites across the northern high latitude permafrost zone, further investigation is warranted on the role of wildfire on permafrost dynamics. We have identified several understudied research areas that could be augmented with this dataset. First, the dataset could be used to further investigate the geospatial distribution of permafrost recovery following fire across the northern high latitude permafrost zone. Second, these data could be used to determine the probability (i.e., likelihood) of permafrost recovery after wildfire as a function of ecotype or ecoclimatic zone, permafrost classification, fire rotation period, and/or climate. Third, the data could aid in determining the soil C consequences of temporary or permanent post-fire permafrost degradation. Fourth, investigations could be structured to identify changes in wildfire activity that affects the likelihood of permafrost recovery/degradation and associated soil C vulnerability using predictive mapping. Fifth, the data could be used to develop an organic layer deficit value that would represent the difference between the organic layer thickness in the burn scar with the organic layer thickness in the unburned control site. Sixth, this dataset could be augmented with quantification of subsidence and the combination of that with ALT to understand how much new permafrost is exposed to seasonal thaw as a result of fire. Finally, there is the opportunity for this dataset to be used in algorithm development, calibration, and validation for evolving process-based models that are trying to capture the impact of fires on permafrost.

**5 Data use guidelines & availability**

The FireALT dataset (Talucci et al., 2024) are publicly available for download through the Arctic Data Center under a Creative Commons Attribution 4.0 International copyright (CC BY 4.0). Data should be appropriately referenced by citing this paper and the dataset (see Arctic Data Center). Users of the data are invited to ask questions by contacting the dataset developers. We recommend that researchers planning to use this data as a core portion of their analysis collaborate with the data developers and relevant individual site contributors. The data are available for download as a csv file through the Arctic Data Center https://doi.org/10.18739/A2RN3092P).

**6 Conclusions**

The FireALT dataset offers a collection of paired burned and unburned plots with measured thaw depths and estimated ALT. By estimating ALT, we address a key challenge: the ability to assess impacts of wildfire on ALT when measurements are taken at various times throughout the thaw season depending on the time of field campaigns (typically June through August). This dataset can be utilised for future research activities that can expand understanding of the feedbacks between permafrost, wildfire, and global climate systems. Changes to the active layer serve as an important diagnostic indicator that requires continuous monitoring under the current dynamic climate conditions to further understand temporary or permanent changes to permafrost and subsequent losses in carbon storage. These types of data synthesis efforts are crucial for addressing understudied research areas particularly algorithm development, calibration, and validation for evolving process-based models as well as extrapolating across space and time, which will elucidate permafrost-wildfire interactions under accelerated warming across the high northern latitude permafrost zone.

**Author contributions**

The FireALT dataset was conceptualised during the 2019 Permafrost Carbon Network meeting by ACT, BMR, DO, KLM, LTB, MAW, MJL, MML with additional input by ACT, AKP, AVR, BMR, JAO, JEH, KLM, LTB, MAW, MJL, MRT, NB, REH, SMN, SV for the methods. Data curation was carried out by AB, ACT, AKP, AS, AVR, BB, BVG, CJFD, CM, CMD, DO, GVF, HDA, JAO, JEH, JLB, KLM, LB, LBS, LRD, LTB, MCM, MML, MRT, MTJ, NB, OS, RAL, REH, SMN, SS, SV, TAD, TAS, TH. Formal analysis was performed by ACT, JEH, MML. ACT and MML provided project management. BMR, MML provided supervision. Visualisations were created by ACT, JEH, JD. ACT, JEH, MML wrote the original draft. All authors contributed to the realisation of the permafrost wildfire data and participated in the editing of the manuscript.

## Competing Interests

S. Veraverbeke is a member of the editorial board of ESSD. The contact author declares that they and all other co-authors have no competing interests.

## Acknowledgments

A. C. Talucci acknowledges Christina Shintani and Greg Fiske at Woodwell Climate Research Center for their cartographic feedback and funding support from the NSF Arctic System Science (award no. 2116864). J. E. Holloway acknowledges Antoni Lewkowicz at the University of Ottawa for the support for field data collections. B.M. Rogers recognizes support from the Gordon and Betty Moore Foundation (grant no. 8414), NSF Arctic System Science (award no. 2116864),  and funding catalysed by the Audacious Project (Permafrost Pathways). A. L. Breen acknowledges Colette Brown, Fernanda Santos, and Thomas Moran for assistance in the field and funding from the Director, Office of Science, Office of Biological and Environmental Research of the US Department of Energy under Contract No. DE-AC02-05CH11231 as part of the Next-Generation Ecosystem Experiments (NGEE Arctic) project. J. O'Donnell acknowledges Jennifer Harden and support from the U.S. Geological Survey for field data collections. D. Olefeldt acknowledges Carolyn Gibson for her field work contributions to the contributed data. L. T. Berner was supported by the NASA Arctic Boreal Vulnerability Experiment (80NSSC22K1244 & 80NSSC22K1247). S.M. Natali acknowledges John Wood and the Polaris Project team for field support, and funding from NSF (1417700, 1915307, 1561437) and NASA (NNX15AT81A). T.A. Douglas acknowledges the U.S. Department of Defense's Strategic Environmental Research and Development Program (Project RC18- 1170) and Environmental Science and Technology Certification Program (Project RC22-D3-7408) as well as the U.S. Army Engineer Research and Development Center Basic Research Portfolio through Program Element PE 0601102A/T14/ST1409. S. Sistla and N. Baillargeon acknowledge support from NSF 2218742. J.L. Baltzer acknowledges funding through the Government of the Northwest Territories' Cumulative Impacts Monitoring Program Project 170, Canada First Research Excellence Fund's Global Water Futures program (project Northern Water Futures), Natural Sciences and Engineering Research Council's Discovery Grant funding, and the Canada Research Chairs program. Data collection was conducted under Aurora Research Institute's Scientific Research License numbers 16815, 16755, 16311, 16018, 15879, and 15609. C. J. F. Delcourt acknowledges funding from the Dutch Research Council (NWO) through a Vidi grant (grant no. 016.Vidi.189.070) and from the European Research Council (ERC) through a Consolidator grant under the European Union's Horizon 2020 research and innovation program (grant no. 101000987), both awarded to S. Veraverbeke. T. A. Shestakova acknowledges funding from  the Beatriu de Pinòs Programme of the Government of Catalonia (2020 BP 00126). K. Manies acknowledges the support of the U.S. Geological Survey Earth Surface Dynamics Program. A.K. Paulson and H. D. Alexander acknowledge Seth Robinson, Eric Borth, Sarah Frankenberg, Aaron Lewis, Brian Izbicki, Clark Thompson, Jill Young, Amanda Ruland, and Elena Forbath for assistance with field work and Valetin Spektor, Nikita Zimov, Sergei Davydov, and Sergei Zimov for contributing extensive knowledge of the region

and logistics support. We also acknowledge NSF OPP-2100773. G. V. Frost acknowledges funding from the Western Alaska Landscape Conservation Cooperative (WALCC) award F16AC01215, NASA Arctic Boreal Vulnerability Experiment contract NNH16CP09C. B.V. Gaglioti acknowledges Park Williams for fieldwork, and NSF Award 2124824 and the Joint Fire Science Program Project 20-2-01-13 for funding. Thanks to Benjamin Maglio and Dana Brown for their assistance in reviewing this manuscript. Thanks to the Arctic Data Center team for their assistance with archiving the dataset. Any use of trade, firm, or product names is for descriptive purposes only and does not imply endorsement by the U.S. Government.

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
