# Peer review of "Permafrost-wildfire interactions: Active layer thickness estimates for"

_Earth System Science Data, 2024_

## Author Response (AR1)

RC1: 'Comment on essd-2024-526', Anonymous Referee #1, 07 Jan 2025

The paper describes the FireALT dataset which includes thaw depth measurements from permafrost sites affected by forest and/or tundra fires and corresponding unburned sites. Although a compiled dataset can provide critical empirical information for assessing the impacts of wildfire on the active layer thickness, in my opinion, the structure of the dataset, the dataset description, and the analysis presented in the paper is rather confusing and require modification before the final publication. Below are specific issues that need to be addressed.

- The paper has a section on "data and methods;" however, there is no information related to specific methods used to measure the thaw depth. Line 157 suggests that "thaw depth… is typically obtained by measuring depth to refusal using a graduated steel probe" and I assume that the data obtained by probing was used exclusively for compiling the dataset, however, there is no indication that this is the case. A more detailed description of the observational methodology is needed. Is there any specific reason why thaw depth observations obtained by other, arguably more accurate methods (e. g. ground temperature, thaw tubes) were not included in the dataset? A more detailed description of the observational methodology is needed.

**AC–Thank you for these suggestions. We will add how each data contributor collected thaw depth measurements to Table 1. Since this is a data synthesis, we did not have control over how contributors collected their thaw depth measurements. We include citations for each data set contributed in Table 1. Finally, we will add text in the methods acknowledging caveats associated with using a probe.**

**"The data sets that contribute to this synthesis were, and is typically obtained by measuring depth to refusal using a graduated steel probe (Brown et al., 2000). A steel probe is a typical means of measurements, however, there is potential for error introduced by issues such as identifying the freeze-thaw boundary, soil variability, subsidence, user bias (Brown et al., 2000, Bonnaventure and Lamoureux, 2013, Strand et al., 2021, Scheer et al., 2023)."**

- Assuming that the probing data was used, what is the rationale behind treating each poke of a probe into the ground as an individual data point? The thaw depth is characterized by extreme spatial variability over very short lateral distances. As a result, a single probing point will not be representative and cannot provide information about thaw depth in the particular area or site. Also, unless each observational point is clearly marked (which usually causes local disturbance and thaw depth alteration) it is impossible to accurately assess the inter/intra annual thaw depth dynamic from repetitive measurements by probing at a single point. This is the reason for spatial thaw depth surveys consisting of multiple thaw depth probing according to a predetermined spatial sampling strategy, used to assess

the thaw depth of a particular location or site. As such an individual probing sample does not have meaning and site-specific statistics should be used for temporal and spatial assessments of thaw depth. The size of the statistical sample and sapling strategy should correspond to the level of spatial variability of local ALT controlled by the variability of surface and edaphic properties. I would assume that all 18 data submissions used such an observational methodology. So why break it apart and treat each poke as an individual data point? Instead, the paper (and the dataset) should include a description of sites, the number of individual probings, and the sampling strategy for each site/year. The analysis related to spatial attributes and comparison between burned and unburned areas should be conducted based on site-specific statistics rather than individual probing measurements. Similarly, the site-specific statistical measures (e.g. mean) should be used to estimate the active layer thickness from early/mid-summer thaw depth measurements. Overall, indicating that a dataset has 48K measurements is highly misleading. That can be especially problematic for the potential use of the dataset for modeling and remote sensing validation as a single measurement is not representative of a grid cell or a pixel. I strongly believe that structuring the dataset and analysis around site-specific statistics can greatly enhance its usability and help avoid potential misinterpretations

**AC–Thank you for these thoughts and suggestions. Our aim with "including each poke" was to be transparent about what data were contributed and how they were processed. Because this is a synthesized dataset, we wanted to be transparent in the data process and recognize that the raw data could potentially be useful for a variety of applications with different needs and various spatial stratifications depending on scale. Incorporating all the data allows users to dive into and describe the variability from points to circumarctic. We will add to the archived data code to aggregate samples into plots, sites, or pairs. Contributed data sets all originated from different studies with different objectives; therefore, spatial sampling designs and methods vary across and within contributed data. We will add a column to Table 1 with a brief description of sampling design for each contributor. Data users can aggregate measurements to whatever level makes sense for their analysis, and we have equipped them with the metadata and data structure to do that. Further, some site/plot level data were collected at different times during the season, making aggregation challenging without standardising to end-of-season ALT, and hence adding an additional layer of processing and assumptions. Ultimately, our goal was to generate a dataset with paired burned/unburned sites, described in section 2.6, while also providing an R script with the archived data that can be used to aggregate the data.**

**To address this comment, we will also add details to section 2.6 about aggregating data to the site/plot level. We will update the archived data with two csv files, one for plot level mean ALT and one for the paired burned-unburned mean ALT. We will also**

update the archive with a script for aggregating site-level data. Please note these updates in the data archive will result in the generation of a new DOI.

We will add summary data of the site/plot level to the abstract and in the results "across 9,432 plots and 388 sites." Additionally, section 3.3 and Figure 7 report based on the aggregated burned/unburned pairs.

- I question the uniform use of a 14-day period to account for the active layer freezing from the bottom for the estimation of the ALT. This period might be reasonable for cold continuous tundra permafrost, but it can be much shorter or non-existent for warm permafrost. I understand that it is impossible to assess this period for all locations, but different approximations should be used for different permafrost zones. These approximations can be obtained from published literature or a few characteristic GTN-P temperature observations.

AC–Thank you for this suggestion. We recognize the limitations of our approach to estimating ALT from raw thaw depth measurements. However, addressing this issue with more detailed approximations for the date of maximum thaw depth is outside the project's scope and would require substantial additional data, analysis, and assumptions. This is one of the reasons we decided to share the raw data and our processing code, which allows for users to edit the approach and for future analyses to further explore these nuances. We will add an additional paragraph to the discussion section 4.2 to address these limitations.

"Similarly, the time at which permafrost begins to refreeze from the bottom varies with permafrost temperature, soil moisture and thermal properties, and local edaphic hydrological conditions. Consequently, our assumption that ALT occurs 14 days before the date at which air temperature drops below freezing is another source of uncertainty. Overall, interannual variability in ALT is dependent on complex interactions between air temperature, precipitation, snow dynamics, hydrothermal processes, water energy exchanges, and fluctuations in thaw season length, which are a source of uncertainty in our approach (Shur et al., 2005, Hu et al., 2023, Grünberg et al., 2024). While in warmer boreal sites the 14 day lag may be longer or non-existent depending on the complex interactions of these landscape-level controls. Despite this, estimating ALT allows for insightful comparisons between sites that are not appropriate or meaningful with the raw data."

- More details related to the assessment of uncertainty are needed. Where did the sample of 626 measurements of seasonal thaw progression come from? Are they from a single or multiple sites? Single or multiple years? Which region?

AC – Thanks for these suggestions and questions. We will add to the description in the paper.

"We used two data sets for this analysis from contributors that had repeat measurements from within a season for early/mid-season and late season at the same locations. These data sets differed as one was a subset of their data

contributed to the data synthesis for the boreal near Yellowknife, Canada (N = 626; Holloway et al. 2024), whereas the other was used solely for quantifying uncertainty for tundra on the Seward Peninsula, AK (N = 37; Breen, unpublished). The tundra data was missing key meta data which precluded it from the synthesis."

- What is the spatial resolution of data used to assign spatial attributes to the dataset? How does it relate to the size and number of the sites (not individual probings) in the dataset?

AC–Thanks for these questions. Three attributes were added through a spatial join. The spatial data was in vector format (i.e., shapefile) and were polygons, not rasters, so we could not provide the spatial resolution. We state in Table 4 that these three variables (resBiome, resName, permaExtent) are generated based on a spatial join. Still, we will add to Table 4 that those attributes are acquired from vector data. Additionally, we will add the following text to section 2.4 about spatial attributes

"The spatial coverage, and hence inherent resolution, of these polygon products is much larger than the data points or any site-level aggregation. Due to the coarser resolution, data contributors designation of biome outweighed what was assigned through the spatial join. The small percentage of plots that were misassigned biomes and ecozones were visually inspected and found to be adjacent to the boundary with the matching biome and were manually reassigned (see code)."

- Finally, I would consider an inability of manual thaw depth probing to capture surface subsidence occurring after the fire as one of the major limitations of the approach and the dataset. The substance is mentioned in lines 397-400 however, it deserves more attention. Since mechanical probing uses a surface as a reference point for observing thaw depth, subsidence can introduce quite a significant bias in thaw depth measurements and assessment of burn/unburned differences in ALT. Moreover, this bias is spatially heterogeneous and temporarily nonlinear. I would think that in ice-rich permafrost this bias can be much greater than uncertainties related to Stefan-based ALT estimates.

AC - Thank you for your thoughts and suggestions. We will add these caveats to the discussion section on limitations. Specifically, we will add the following after the sentence that ends on line 400.

"Subsidence is not accounted for in the synthesised data. Subsidence can introduce additional bias in the measurement of ALT since thaw depth probing uses the surface as a reference. In areas where subsidence had occurred after fire, our data set will underestimate the magnitude of active layer thickening caused by fire. Bias from subsidence is difficult to estimate because it would be spatially heterogeneous, temporarily nonlinear, and largely dependent on ice content (Shiklomanov et al., 2010, O'Neill et al., 2023, Painter et al., 2023)."

RC2: , Anonymous Referee #2, 07 Jan 2025

Anna C. Talucci et al.'s manuscript, "Permafrost-wildfire interactions: Active layer thickness estimates for paired burned and unburned sites in northern high-latitudes," classically orders an important characteristic of northern permafrost regions: active layer thickness (ALT) during wildfires. With global warming, permafrost-wildfire interactions are becoming a more significant problem. The Permafrost-wildfire interactions system's ALT database contains 47,952 ALT estimates (27,747 burned, 20,205 unburned) across 32 attributes. 193 unique paired burned/unburned sites are distributed across 12 ecozones in Canada, Russia, and the United States. This data covers fire events between 1900 and 2022. This is in line with the main goal of the ESSD journal.

The competent compilation of ALT distribution data in paired burned/unburned sites considers calculation of single measurements of thawing depth to the maximum thickness of thawing depth by the end of the season, known as ALT. The ALT calculations were clearly presented by the authors. The main calculations were performed in ecozones.

Major comments:

1) I felt the authors could have more clearly articulated the study's objective in the Introduction. The authors' intentions remain unclear. Consequently, the reader may easily become confused. The ALT analysis is applicable to both ecozones (broad generalizations) and landscapes (detailed generalizations). Please clearly justify your decision.

**AC–Thank you for this suggestion. We will clarify the objectives by identifying five main objectives in the last paragraph of the introduction and specify each see text below. Additionally, the ecozones were added to the data set to provide additional context as we think they are a useful spatial attribute. While we provide a map and some summary data using the ecozones the main calculations are with the raw data or the paired burned/unburned means.**

**"The overarching goal is to generate a synthesised data set of ALT for burned/unburned pairs. To achieve this, we had four main objectives for the paper: 1) describe how the data was collected and synthesised for thaw depth measurements of burned sites with paired unburned sites, 2) describe how we standardised thaw depth measurements to end-of-season ALT with estimates of uncertainty,  3) provide details on how to aggregate data to plot, site, and paired burned/unburned means and provide a summary of the data set, and 4) discuss the strengths and limitations of the dataset, along with its potential uses."**

2) Given the uncertainties of ALT analysis caused by varying landscapes, a separate subsection discussing this issue is requested. Showing how ALT differs based on soil and vegetation type would highlight ALT uncertainty. This demonstrates your analysis of regional

characteristics using an ecozone approach. There are other problems you haven't addressed, including those at the landscape level.

**AC– Thank you for these suggestions. We will add to the discussion. Specifically, we will add a paragraph at the end of section 4.2 addressing these items and the requests in item 3 below.**

**"In addition to these physical controls, there are additional biogeomorphic factors that influence changes in ALT from fire. Landscape scale variation in topography, soil type and moisture, ground ice content, and vegetation cover and regrowth are all sources of uncertainty (Shiklomanov et al., 2010, O'Neill et al., 2023, Painter et al., 2023) accounting for these drivers would require datasets that may or may not be available, and is a separate research effort outside the scope of this paper. We use ecozones to highlight summary statistics of the data set since ecozones are characterised by sharing similar climates, geologic substrates, vegetation, and landforms. The use of ecozones for providing a broad overview of the data, which captures some of the variability in ALT measurements; however, finer-scale landscape features likely still add substantial variation to the estimated ALT and changes from fire. Future work could analyze how microtopographic features that influence local hydrology, burn severity, vegetation structure and function, and ice content impact wildfire-induced changes in ALT. "**

3) The variations in TDD across different time periods are significant. Considering permafrost-wildfire interactions from 1900-2022, climate change significantly alters their effects. Modern warming's onset spurred cryogenic process development. Therefore, this point, maybe along with point 2 of my comment, should be reflected in the text.

**AC – Thanks for these suggestions. We will add to the additional paragraph mentioned above the following sentences.**

**"Further, while growing season lengths and thawing degree days have increased over the last century (e.g., Barichivich et al., 2012), the data synthesised here was only measured from 2001 onward despite covering fire events from 1900-2022. Recent thaw depth measurements from areas that burned more than several decades ago represent a post-fire evolution of the active layer under climatic conditions that no longer exist. The snapshot of thaw depth related to wildfire events in space and time provided by this data set may therefore include climatic effects that are hard to disentangle. Warming trends in the northern high-latitudes have influenced cryoturbation (e.g., Liu et al., 2024), which may bias the estimated ALT."**

4) The Introduction of the manuscript required citing the significant contributions of Moskalenko (1999, 2012) on Northwest Siberia, Kirdyanov (2020) on Central Siberia, Lytkina (2008), and Petrov (2022) on Central Yakutia.

**AC–Thanks for these suggestions. We have tracked down the following citations based on the information you provided. Unfortunately, we have been unable to acquire Moskalenko (2012); however, we have found some additional publications that focus on parts of Eurasia that we will integrate in addition to Moskalenko (1999), Lytkina (2008), Kirdyanov (2020) and Petrov (2022). We will incorporate citations for the papers acquired into the introduction lines 107-110,**

**"Similarly, tree canopy removal reduces shading in the summer and results in more snow on the ground in the winter, both leading to higher surface soil temperatures and expansion of the active layer into near-surface permafrost, which has been shown across North America (Rocha et al., 2012, Jafarov et al., 2013, Jiang et al., 2015, Zhang et al., 2015, Douglas et al., 2016, Fisher et al., 2016, Gibson et al., 2018) and Eurasia (Moskalenko 1999, Lytkina 2008, Kirdyanov et al., 2020, Heim et al., 2021, Fedorov, 2022, Petrov et al., 2022)."**

**Additionally, we will integrate Chebykina et al., 2022 and Moskalenko 1999 into the sentence that starts on line 102**

**"Biomass combustion during fires removes the insulating surface vegetation (i.e., moss, lichen, low growing shrubs) and soil organic matter, typically reduces evapotranspiration (Rouse 1976, Amiro 2001, Chambers and Chapin 2002, Chambers et al., 2005, Amiro et al., 2006, Chebykina et al., 2022, Fedorov, 2022), and reduces short-term albedo during the thaw season, resulting in increases in the ground heat flux and the expansion of the active layer (Moskalenko 1999, Rocha et al., 2012, Jafarov et al., 2013, Nossov et al., 2013, Jiang et al., 2015, Douglas et al., 2016, Fisher et al., 2016, Gibson et al., 2018)."**

**Finally, we will also add Moskalenko 1999 to the sentence that starts on line 109,**

**"In contrast, across North American Arctic tundra, shrub removal from wildfire results in thinner snow due to increased wind exposure, which causes a reduction of the active layer (Wang et al., 2012, Jones et al., 2024), while Russian scientists note an expansion of the seasonal active layer that is dependent on vegetation communities (Moskalenko 1999, Lytkina 2008)."**

Specific comments:

1) The solid and dotted lines in graphs A and B should be indicated in the caption of Fig. 4. **AC–Thanks for catching this. We will add "solid line for burned distribution and dashed line for unburned distribution" to the figure caption.**

2) The abscissa axis in Fig. 6 C displays 16 points (sections?). Only unburned sections are given in general terms in the captions. What is the nature of these sections? Why aren't they included here and in the supplementary materials?

**AC–We will add the following text to the caption to clarify, "Burned sites include b1, b12, b13, b14, b15, b16, b5, b9, DR, GC, ML, MP85, and MP86, and unburned sites are ub2, ub3, and UbMp80."**

In general, I believe that the manuscript is a source of new knowledge. From my perspective, there is a need for improvement in their current form. In my opinion, the authors are capable of doing it.